# Transformers from Compressed Representations

## Abstract

Compressed file formats are the corner stone of efficient data storage and transmission, yet their potential for representation learning remains largely underexplored. We introduce **TEMPEST** (TransformErs froM comPressed rEpreSenTations), a method that exploits the inherent byte-stream structure of compressed files to design an effective tokenization and encoding strategy. By leveraging this compact encoding, a standard transformer can directly learn semantic representations from compressed data streams, bypassing the need for raw byte-level processing or full media decoding. Our proposal substantially reduces the number of tokens required for semantic classification, thereby lowering both computational complexity and memory usage. Through extensive experiments across diverse datasets, coding schemes, and modalities, we show that TEMPEST achieves accuracy competitive wit the state-of-the-art while delivering efficiency gains in memory and compute.

## 1 Introduction

Transformer architectures were originally developed for language processing Vaswani et al. (2017) and later extended to various multimedia domains (*e.g.*, images, audio, video Dosovitskiy et al. (2020); Gong et al. (2021); Arnab et al. (2021)). A major challenge in applying transformers to such modalities is that a few seconds of audio or video can yield very long tokenized sequences Arnab et al. (2021); Tay et al. (2020). Such long sequences pose a key limitation, as the memory and computational requirements of the attention mechanism scale quadratically with the sequence length. Prior work has attempted to mitigate this issue through approximate attention Choromanski et al. (2021); Xiong et al. (2021), reduced-complexity operations Wang et al. (2020); Child et al. (2019), or token merging strategies Haurum et al. (2023); Bolya et al. (2022). In this paper, we address the complexity issue of attention-based architectures from an orthogonal direction: we remap the input data into shorter, compact sequence that preserves the relevant features with minimal information loss.

Our proposal leverages existing entropy coding schemes developed for multimedia, namely Compressed File Formats (CFFs). CFFs provide a compact representation of their original data, and offer the advantage of efficient storage and transmission while retaining all meaningful perceptual information. This property has recently attracted research interest Pérez et al. (2024); Yu et al. (2023); Wu et al. (2024b); Horton et al. (2023b), as compact data representations (i.e., data with shorter sequences) can potentially mitigate the large memory requirements of modern attention-based models. Although the work of Pérez et al. (2024) showed that a transformer decoder architecture can learn useful features from a compressed data stream. Their approach provided no empirical benefit in terms of memory complexity, network size or training time.

In this paper, we advance representation learning from compressed file formats (CFFs) and show that byte-level attention is largely unnecessary when working with compressed data streams. We observe that many CFFs already incorporate an inherent notion of sub-components within the format. While these sub-components are primarily designed to enable robust error handling and on-the-fly decoding independent of preceding or subsequent segments, we demonstrate that they also provide a natural tokenization scheme. Specifically, each sub-component can be independently embedded using a lightweight transformer network, yielding a shorter token sequence. This tokenized sequence can serve as the input to a second transformer architecture Vaswani et al. (2017), which learns semantic

representations over it. Importantly, this second transformer operates on much shorter sequences compared to models trained directly on the raw byte streams.

Our approach leverages the compactness and structured nature of compressed data in CFFs and their sub-components, and enables direct semantic modelling across diverse compressed data formats while substantially reducing the number of tokens per sub-component. As a consequence, we lower both the computational complexity and memory requirements in the final model. Overall, our method benefits from then compact coding in CFFs, which inherently remove the redundancy and irrelevant data while fully preserving semantics at the cost of minimal perceptual loss.

We name our method TEMPEST (TransformErs froM comPressed rEpreSenTations). We empirically demonstrate that TEMPEST achieves results competitive with the state-of-the-art models across various datasets, low-entropy coding schemes and modalities, while also offering empirical memory savings. Our contributions are three-fold: i) We introduce the first practical tokenizer for compressed data, our proposal that can work across multiple CFF and multiple modalities. ii) We show that a standard transformer encoder architecture can learn semantic representations from the compressed data tokenizer, resulting in competitive results with state-of-the-art audio classification methods. iii) We show that, after tokenization, we have a small token sequence that results in empirical memory savings.

To ensure reproducible results and foster future research, we will made all the resources of this project available upon acceptance. These resources include official weights, training code and benchmark results.

## 2 RELATED WORK

**Byte Sequence Modeling for Uncompressed Data.** Recent advances in byte sequence modeling have introduced techniques to handle the challenges of raw data streams. The MegaByte framework Yu et al. (2024), for example, employs a multi-scale decoder transformer architecture and introduces a "byte patchification" strategy to manage the length and complexity of byte sequences. Similarly, bGPT Wu et al. (2024a) adopts a decoder-only Transformer for autoregressive byte sequence generation. While these approaches advance the modelling of byte-level data, they remain limited to uncompressed inputs and overlook the more prevalent and practical case of compressed file formats. In contrast, our work directly addresses this format gap by operating on the compressed byte stream, using MP3 encoding as a primary case study.

**Training over partially decoded JPEGs.** Rather than relying on raw RGB pixel values, some related studies have proposed Convolutional Neural Network (CNN) Gueguen et al. (2018); Verma et al. (2018) and Vision Transformer (ViT) Park & Johnson (2023) architectures that operate directly on partially decoded JPEG representations. For example, Gueguen et al. (2018) introduced a CNN LeCun et al. (1995) capable of processing such inputs, enabling faster image handling by bypassing parts of the decoding pipeline. In a similar spirit, Park & Johnson (2023) demonstrated that a ViT Dosovitskiy et al. (2020) can effectively leverage partially decoded JPEGs, allowing the model to engage with data in a more compressed form. Together, these works show an initial shift towards exploiting the compactness and some structural characteristics of compressed formats, while still retaining some degree of decoding. In comparison our proposal fully relies on the sub-component structure of the CFF, moreover, no decompression (not even partial) is performed at any stage of our pipeline

**Direct Operations on Compressed JPEG Byte Sequences.** ByteFormer Horton et al. (2023a) explored the processing of compressed formats at the byte level, demonstrating that JPEG byte sequences present substantial challenges due to their non-linear encoding and variable length. The study further showed that conventional byte patching strategies can hinder performance, given the high information density of compressed data. The work of Pérez et al. (2024) is perhaps the closest approach to our method, however it is more focused on byte-level modelling and error handling. This method incurrs in a large performance penalty given that the length of the sequence is never reduced. In contrast, our approach formulates a novel byte tokenisation strategy that reduce significantly the size of CFFs sequences, moreover, our design operates on significantly less tokes per CFF component thus reducing the overall computational complexity of the method.

## 3 Transformers from Compressed Representations

Bytes constitute the natural vocabulary of compressed file formats (CFFs), with a fixed alphabet of 256 possible values (e.g., `0x00`, ..., `0xFF`). However, compression algorithms often operate at the bit level, packing information across byte boundaries. As a result, a single byte is not always self-contained: its bits may simultaneously contribute to multiple elements of the compressed stream. This implies that bytes lack intrinsic semantic meaning. We argue that these two aspects (i) sub-byte encodings and (ii) the absence of semantic self-containment, pose fundamental limitations for deep models that rely on byte-level tokenization.

To overcome this limitation, we leverage the structural design of many compression schemes (e.g., MP3, Opus, JPEG), which are organized around *blocks*. A block represents the smallest encoded unit that can be decoded independently of the rest of the stream. By construction, blocks encapsulate self-contained information, making them a more suitable unit for tokenization than individual bytes. Crucially, each block is encoded independently, and all blocks follow the same compression scheme, this inter-block consistency enables deep models to generalize across them while learning from individual blocks which constitute relatively short data sequences.

Building on this insight, we treat blocks as the atomic units for tokenization in CFFs, our first goal is to learn to recover useful feature embeddings directly from each individual compressed block. Then we use use these block-level features in a standard transformer architecture which aggregates the information across the sequence of blocks found within a single CFF. This design allows us to exploit the inherent structure of compressed formats while avoiding the inefficiencies and ambiguities of byte-level modelling. Building on this insight, we treat blocks as the atomic units for tokenization. Our first goal is to recover useful feature embeddings directly from each individual compressed block. These embeddings are then aggregated by a standard transformer encoder across the multiple blocks within a single CFF.

At its core, TEMPEST combines a novel token embedding scheme designed for compressed representations with a vanilla transformer encoder Vaswani et al. (2017), which aggregates features across sequences of embedded blocks. By jointly optimizing the tokenization and aggregation stages, TEMPEST produces a semantic embedding that captures the full content of the compressed data stream and enables tasks such as classification. These components are illustrated in figure 1.

### 3.1 Embedding Compressed Byte-streams

The key insight of the embedding scheme in TEMPEST is that most compressed representations inherently define sub-components within their encoding format. We exploit this property by defining tokens according to the blocking strategy specified in each compression standard, and map each compressed block to one or a few tokens. For instance, the MP3 iso (1993) standard organizes data into frames, JPEG itu (1992) defines Minimum Coded Units (MCUs) as local blocks, and Opus Valin et al. (2012) encodes audio as sequences of frames contained within packets.

**Block Embedding.** Let $S = \{s_0, s_1, \ldots, s_{n-1}\}$ denote a compressed byte stream of length $n$, where each $s_j$ represents a single byte in $\{0, \ldots, 255\}$. We treat $s_j$ as the integer representation of the underlying 8-bit value (i.e., $0 \equiv$ `0x00`, $255 \equiv$ `0xFF`). Many compression standards is that the stream is organized into *blocks*, these blocks are delineated by format-specific markers or headers, which allow us to identify their byte boundaries. Using these markers, we partition $S$ into $i \ll n$ blocks:
$$S = \{B_0, B_1, \ldots, B_{i-1}\}, \quad B_k = \{s_m, \ldots, s_{m+L-1}\},$$
where $L$ denotes the length (in bytes) of block $B_k$ which starts at byte $m$. For simplicity, if a Blocks shorter than $L$ we pad with a special tokens which are outside of the vocabulary of the compressed representation (e.g., integer value 256).

The definition of the CFF and the partitioning of the compressed stream around the byte markers, ensures that the semantic information encoded by the compression algorithm is preserved within each block $B_k$ *i.e* no spurious or incomplete information from a contiguous block is added, no information is leaked into another block. As outlined earlier, these blocks reflect meaningful units of the underlying compression scheme (e.g., an MP3 frame corresponds to a fixed-duration audio segment, while a JPEG MCU encodes a local spatial region of an image).

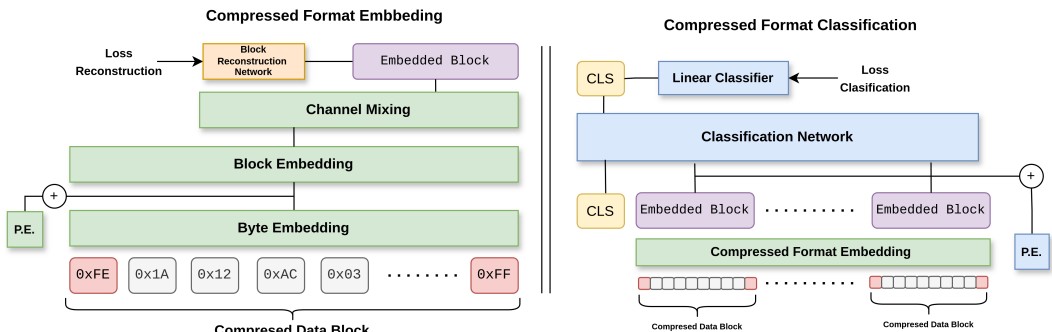

Figure 1: **TEMPEST Architecture.** The proposed model consists of three sub-networks: the block embedding network (green), the classification network (blue), and the block reconstruction network (orange). The input to TEMPEST is a compressed byte stream (gray), which is split into sub-components (compressed data blocks) according to the special byte markers defined in the CFF (red, the byte values shown are only for illustration). Each compressed block is mapped to an embedding (purple), which is regularized by the reconstruction network. The classification network is ViT-like: it prepends a [CLS] token (yellow) and produces the final classification from the sequence of compressed embeddings.

After extracting the block sequence $B_k$ from the CFF stream, we embedded them independently. First, each byte is mapped into an $N$-dimensional vector space, producing $B_k \in \mathbb{R}^{L \times N}$. Each sequence is then processed independently by a lightweight stack of transformer encoder layers $E_t$ which operates in an intra-block attention scheme. Following this, a channel-mixing network $M$, inspired by the MLP-Mixer Tolstikhin et al. (2021), projects the embeddings reducing the sequence length from $L$ to $L'$ (with $L' << L$). The results is compact, high-dimensional representation of $B_k$ with length $L'$, we name this sequence the *Block Embedded Sequence ($\mathcal{B}$)*:

$$\mathcal{B}_k = M(E_t(B_k)), \quad \mathcal{B}_k \in \mathbb{R}^{L' \times N}.$$

To ensure that the lock Embedded Sequence remains informative, we regularize it with a reconstruction objective, effectively casting the block embedding process similar to an auto-encoder Hinton & Salakhutdinov (2006); Goodfellow et al. (2016). Specifically, a $P$-layer transformer decoder $D_t$ is trained to regress the original byte sequence:

$$\hat{B}_k = D_t(\mathcal{B}_k),$$

with a 256-way classification (255 unique bytes and the padding byte) head and cross-entropy loss applied at each position, treating each byte as a categorical variable. Unlike a conventional autoencoders, our goal is not perfect reconstruction of the original signal, but rather to encourage $\mathcal{B}_k$ to serve as both (i) a semantically meaningful representation and (ii) a compact approximation of the original compressed block.

**Learning From Block Embedded Sequences.** Once the lock Embedded Sequence $\mathcal{B}$ has been estimated, we work towards building a to semantic representation from $\mathcal{B}$. Our approach follows the design of a Vision Transformer (ViT) Dosovitskiy et al. (2020), where the elements of $\mathcal{B}$ are treated as tokens analogous to visual patches (i.e., each token corresponds to a contiguous segment of the original data, already mapped into a compact embedding by the block embedding network). We apply a sinusoidal positional encoding to $\{\mathcal{B}_0, \mathcal{B}_1, \ldots, \mathcal{B}_{i-1}\}$ and prepend a learnable [CLS] token. The sequence is processed by a transformer encoder $E_c$, and the [CLS] output is mapped by a linear classifier $W$ to the predicted class $\hat{c}$:

$$\hat{c} = W(E_c(\mathcal{B})).$$

Overall, TEMPEST consists of three sub-networks: (i) the *block embedding network* $M \circ E_t$, (ii) the *block reconstruction network* $D_t$, and (iii) the *classification network* $W \circ E_c$. These component are trained jointly with two objectives: a reconstruction loss $L_r$, and a classification loss $L_c$. The final training objective can be expressed as:

$$L = L_r + \lambda L_c = \mathcal{L}(\hat{B}_k, B_k) + \lambda \mathcal{L}(\hat{c}, c)$$

## 4 EXPERIMENTS

We evaluate TEMPEST across both audio and image domains to assess its effectiveness and generality. Specifically, we consider two widely used audio compression formats, MP3 and Opus, and one image compression format, JPEG. For each case, we provide implementation details, training setup, and evaluation protocols, followed by a discussion of the results.

**Compressed Audio.** We first evaluate TEMPEST on audio streams, focusing on the MP3 and Opus formats. MP3 is based on perceptual audio coding, where the block-level units are referred to as frames. Frame size in MP3 depends on both the bit rate and the sampling rate; in our setup, we fix the sampling rate at $44$ kHz and vary the bit rate between $20 - 32$ kbps. Under these settings, frames are nearly uniform in length with an average of $L = 144$ bytes, corresponding to approximately 35 milliseconds of audio. Whenever a shorter frame occurs, we pad it with the special token value of 256. For Opus, we use a fixed bit rate of 29 kbps, which also produces frames of $L = 144$ bytes. Unlike MP3, we do not consider multiple bit rates for Opus in this study.

An important property of compressed audio is that varying the bit rate implicitly serves as a form of data augmentation. For instance, MP3 streams generated at 20 kbps and 32 kbps differ in approximately $90\%$ of their byte values for the same underlying audio signal. While the decoded waveforms are not identical, the differences are designed to be perceptually negligible to human listeners. In contrast, the compressed bytestreams differ substantially across bit rates. We show that TEMPEST is robust to such variability and can even benefit from it, leveraging the diversity in the compressed representations to improve generalization.

**Compressed Images.** We follow the a similar methodology for images. However extracting MCUs from JPEG files is more complex as there is not an exact byte marker between MCUs. Rather, a scan-line (a set of consecutive MCUs which align left to right and cover an $8 \times n$ horizontal region image where $n$ is the image width) of MCUs is encoded together. Nevertheless we approximate the boundaries of MCUs over JPEG files encoded with a quality setting of 75. In the JPEG CFF, the MCUs have variable sizes, despite the miss-alignments between compressed audio and image data, we show that TEMPEST is robust to variable block sizes, and can still operate over inexact block boundaries.

**Implementation Details.** We implement TEMPEST in PyTorch Paszke et al. (2019) using the Accelerate library Gugger et al. (2022). The same overall architecture is employed across all datasets and formats, with the exception of varying the number of layers in the reconstruction and classification networks. Both networks share the same dimensionality for the Key, Query, and Value projections, which we fix at 216, and use feed-forward layers of dimension 864. We set $L' = 1$ (i.e., a single token per block $\mathcal{B}_k$) and weight the reconstruction and classification losses equally ($\lambda = 1.0$). Unless otherwise stated, all reported TEMPEST models contain 5.8M parameters.

For audio experiments, we adopt a data augmentation regime inspired by Trivial Augment Müller & Hutter (2021) and CutMix Yun et al. (2019). Trivial Augment is applied directly to the raw audio waveform, which is subsequently transcoded into a noisy MP3 stream. CutMix is applied to both audio and image data: a subset of compressed blocks is replaced with blocks at the same starting and ending positions from another element in the batch, while the corresponding labels are linearly blended, enabling soft class assignments during training.

For ESC-50 and SC2, models are trained on entire audio clips. For AudioSet, we employ a two-stage training strategy: pretraining on 2-second clips followed by fine-tuning on 4-second clips.

### 4.1 DATASETS

**ESC-50.** ESC-50 Piczak (2015) is a dataset of environmental audio recordings spanning 50 classes such as animal sounds, human activities, natural phenomena, and background noises. It contains a total of 2,000 clips, each 5 seconds long, recorded at 44.1 kHz. The dataset is evenly balanced across classes, with 40 examples per class. Following standard protocol, we report results using 5-fold cross-validation, ensuring that clips from the same fold are held out during training. This setup allows for robust comparison with prior work on ESC-50.

**Speech Commands v2 (SC2).** The Speech Commands v2 dataset Warden (2018) is designed for keyword recognition. It consists of 105,829 one-second audio clips of 35 spoken words, recorded from thousands of speakers at 16 kHz. The official dataset split contains 85,511 clips for training, 10,102 for validation, and 10,489 for testing. Each example is short and homogeneous in length, making the dataset well-suited for small-footprint audio classification models.

**AudioSet.** AudioSet Gemmeke et al. (2017) is a large-scale dataset of 10-second YouTube clips labeled with a hierarchical ontology of 527 sound event classes. It contains approximately 2.1 million annotated clips with a highly imbalanced class distribution. The average clip duration is 10 seconds sampled at 16 kHz. In our experiments, we train only on the balanced subset of 20,371 clips, and evaluate on the official evaluation split.

**MNIST.** MNIST LeCun et al. (2002) contains 70,000 grayscale images of handwritten digit across 10 classes (digits 0–9). Each image has a resolution of $28 \times 28$ pixels. The dataset is split into 60,000 training and 10,000 test images, with a balanced distribution across classes.

## 4.2 Comparison against state-of-the-art

We begin by evaluating TEMPEST classification accuracy against its most direct baseline, the Audio-spectrogram transformer (AST) Gong et al. (2021). Similar to AST, our method uses a standard transformer encoder and a `[CLS]` token for classification. Their only difference lies in the input data, while AST uses a patchified spectrogram as the transformer input, TEMPEST uses the raw compressed stream of the audio file. Due to computational constraints, we are unable to perform large-scale pretraining, therefore, we compare TEMPEST and AST accuracy when both models are trained from scratch. We use the accuracy metrics reported in Gong et al. (2022) for both methods. Table 1 summarizes our results.

Table 1: **Results on audio datasets.** We evaluate TEMPEST against the baseline of ASTGong et al. (2021) in the Speech Commands V2, ESC50 and AudioSet datasets. We show competitive results when learning from scratch. In addition, we report the number of *Tokens Per Second* in the model (TPS), TEMPEST shows a significant reduction in the sequence lenght per second

| Method | AudioSet ↑ | ESC-50 ↑ | SC2 ↑ | TPS ↓ | FLOPS ↓ |
|--------|-----------|----------|-------|-------|---------|
| AST | **14.80** | 41.90 | **92.60** | 108 | 25.65 G |
| TEMPEST | 14.44 | **58.98** | 91.15 | **32** | **16.85 G** |

Overall we observe that TEMPEST can improve over AST when it is applied in data-limited datasets which is the case of ESC-50. For larger datasets, our method remains close to the baseline performance of AST despite having a significantly shorter token sequence. We also measure the length of the token sequence with the number of *Tokens Per Second* (TPS), in comparison TEMPEST only adds 32 tokens for each second of audio, meanwhile AST adds 108 tokens per second. We note that the memory footprint of some operations is drastically improved due to our tokenization scheme which results in a reduced sequence length. In particular for a single second of audio the attention matrix in TEMPEST is about an order of magnitude smaller than in AST (11664 total elements in AST vs 1024 elements in our proposal).

## 4.3 Ablations

We conduct a series of ablation studies to evaluate the impact of key architectural design choices in TEMPEST. Specifically, we focus on three critical components: (i) $L'$, the length of the Block Embedded Sequence ($\mathcal{B}_k$), which directly affects memory usage and is expected to influence downstream performance, (ii) the contribution of the reconstruction loss ($L_r$), which serves as a regularizer for the block embedding network, and (iii) the number of encoder layers ($P$) used in the block embedding network $D_t$, which controls the model capacity at the block level.

**Length of Block Embedded Sequence** ($L'$). As described in Section 3.1, the block embedding process reduces the sequence length from $L$ to $L' \geq 1$ through the channel-mixing transformation. We study the effect of varying $L'$ in a range $1 \leq L' \leq 4$ and report its impact on classification

accuracy. In addition to accuracy, we also measure the *Token-to-Byte Ratio* (TBR), defined as the number of embedded tokens relative to the number of bytes in the original compressed block ($L'/L$).

Table 2: **Ablation on the Block Embedded Sequence Length** ($L'$). Classification accuracy of TEMPEST as a function of the number of tokens $L'$ in the Block Embedded Sequence $\mathcal{B}$.

| Length of $\mathcal{B}_k$ | ESC-50 ↑ | SC2 ↑ | TPS | TBR |
|---|---|---|---|---|
| 1 Token | 58.98 | 91.15 | 32 | 1/144 |
| 2 Token | 56.08 | 91.27 | 64 | 2/144 |
| 3 Token | **59.29** | **91.92** | 96 | 3/144 |
| 4 Token | 55.75 | 91.05 | 128 | 4/144 |

We observe in Table 2 that increasing $L'$ from one to three tokens leads to consistent improvements in accuracy, with the best results obtained at $L' = 3$ on both ESC-50 and SC2. This suggests that using multiple tokens per block enables the model to capture richer intra-block structure. However, further increasing to four tokens results in a decline in performance. Overall, while $L' = 3$ achieves the highest accuracy, the relative gains over the single-token baseline are below 1%, indicating that $L' = 1$ provides a more favorable accuracy–efficiency trade-off.

**Reconstruction Loss $L_r$.** We investigate the contribution of the reconstruction loss $L_r$ as a regularizer. As expected, the main driver of performance is the supervised classification loss $L_c$. However, adding $L_r$ provides consistent improvements on both ESC-50 and SC2, indicating that enforcing reconstruction of the compressed blocks helps the model learn more informative embeddings.

Table 3: **Ablation by loss functions.** The main driver of performance is the Classification loss $L_c$. However the reconstruction loss provides an empirical improvement in both the ESC-50 and SC2.

| Loss Setting | ESC-50 | SC2 |
|---|---|---|
| $L_c$ | 57.38 | 90.37 |
| $L_c + L_r$ | **58.98** | **91.27** |

Table 3 shows that including $L_r$ improves accuracy from 57.38 to 58.98 on ESC-50 (+1.60 absolute, +2.8% relative) and from 90.37 to 91.27 on SC2 (+0.90 absolute, +1.0% relative). These modest but consistent gains indicate that adding the reconstruction loss is beneficial.

**Depth of Block Embedding Network $E_t$.** We next analyze the effect of varying the number of encoder layers in the block embedding network $E_t$, jointly adjusting the classification network depth so that the total number of layers remains constant. Results are shown in Table 4. On SC2, increasing the depth of $E_t$ from one to three layers yields a modest accuracy improvement, with the best performance obtained at three layers (91.59). This suggests that adding capacity to the block-level encoder allows the model to extract richer features from compressed block. However, deeper embedding networks also incur higher computational cost, since they operate on the full frame sequence of 144 tokens, whereas the classification network processes a shorter sequence of only 32 tokens. For this reason, we adopt $E_t = 2$ as a practical compromise between accuracy and efficiency.

Table 4: **Ablation by Network Depth.** We observe improved performance by using deeper encoder networks, although the network have the same number of total layers. Encoder layers are more computationally heavy, and the improves performance comes at the cost of computational performance.

| Embedding | Classification | ESC-50 ↑ | SC2 ↑ | FLOPs ↓ |
|---|---|---|---|---|
| 1 | 8 | 54.92 | 91.31 | **11.92 G** |
| 2 | 7 | 58.98 | 91.27 | 16.85G |
| 3 | 6 | **59.64** | **91.59** | 21.78 G |

**Training with Multiple Bit Rates.** As discussed in Section 3, compressed audio streams generated at different bit rates can differ substantially at the byte level, even when the underlying waveform is perceptually similar. To assess whether leveraging this variability is beneficial, we train TEMPEST using either a single bit rate (32 kbps) or a mixture of bit rates (20, 26, and 32 kbps). During multi-bit rate training, the bit rate is sampled uniformly at random for each training example.

Table 5: **Training with multiple bit Rates.** We empirical observe that including multiple bit rate at training time represent results in an improve performance. For CFF the coding rate acts as data augmentation.

| Training Bit Rates | ESC-50 ↑ | SC2 ↑ |
|---|---|---|
| 32 kbps | 56.66 | 91.06 |
| 20, 26, 32 kbps | **58.98** | **91.27** |

Results are reported in Table 5. On ESC-50, multi-bit rate training improves accuracy from 56.66 to 58.98 (+2.32 absolute, +4.1% relative). On SC2, we observe a smaller but consistent improvement, from 91.06 to 91.27 (+0.21 absolute, +0.2% relative). These results demonstrate that exposing the model to compressed representations at different bit rates acts as an effective form of data augmentation, improving accuracy and generalization.

**Inference With Multiple Bit Rates.** We observe that training with multiple bit rates serves as a form of data augmentation. Building on this property of CFFs, we further investigate whether performing inference at multiple bit rates for the same input can improve accuracy. In this setting, an audio file is re-encoded at different bit rates, and the corresponding predictions are aggregated at inference time. As shown in Table 6, TEMPEST empirically benefits from the complementary information preserved across bit rates, achieving higher accuracy than using a single encoding. This effect is analogous to multi-crop evaluation in vision tasks, where multiple views of the same input provide additional robustness. Our results indicate that multi-bitrate inference offers a simple yet effective strategy for improving compressed-domain audio classification.

**Trivial Byte Embedding.** We conclude our ablation studies by discarding the block-level analysis and instead embedding each compressed byte directly as a token. In this configuration, all components of TEMPEST except the classification network are removed, and the raw byte stream is fed into the model. Despite the relatively small size of the network, the extreme sequence length made it infeasible to train on full inputs, as even a single-sample batch could not fit on an A100-80GB GPU. To enable comparison, we restricted the input to 14 frames (approximately 0.41 seconds of audio) and trained on SC2, achieving an accuracy of 7.81%. While this is above random chance, it remains far below both our proposed approach and standard baselines such as AST, highlighting the inefficiency of naive byte-level tokenization.

## 4.4 ADDITIONAL CFF FORMATS

We conclude our empirical evaluation with a small study of TEMPEST on two additional compressed file formats: Opus for audio and JPEG for images.

Table 6: **Inference with multiple bit rates.** Accuracy on SC2 increases when combining progressively more bit rates, suggesting complementary information across rates.

| Inference Bit Rates (kbps) | SC2 |
|---|---|
| 23 kbps | 91.17 |
| 32 kbps | 91.27 |
| 23, 26 kbps | 91.33 |
| 23, 26, 29 kbps | 92.07 |
| 23, 26, 29, 32 kbps | **92.32** |

Table 7: **Results for the Opus encoding.** We find an average lower performance for the Opus encoding when compared to MP3. Hoevere it remains above the AST Baseline

|  | Split 1 | Split 2 | Split 3 | Split 4 | Split 5 | Average |
|---|---|---|---|---|---|---|
| MP3 | 56.52 | 57.06 | 58.96 | 58.69 | 63.68 | 58.98 |
| Opus | 48.69 | 49.42 | 51.51 | 45.45 | 51.76 | 49.36 |
| AST | - | - | - | - | - | 41.90 |

**Opus Encoding.** The Opus codec follows a frame-based structure similar to MP3 but employs a completely different byte-level encoding mechanism. Table 7 reports results on ESC-50 using Opus encoding compared to MP3. While Opus underperforms relative to MP3, the average accuracy across the five folds (49.36%) remains above the AST baseline. We emphasize that for Opus we tuned only the learning rate of the embedding network and restricted training to a single bit rate. Further improvements may be possible by tailoring the Block Embedded Sequence design specifically for Opus.

**JPEG Encoding.** The JPEG format poses additional challenges compared to MP3 or Opus. In particular, JPEG does not define explicit byte sequences to delimit the boundaries of Minimum Coded Units (MCUs). Instead, these markers are embedded in sub-byte representations, requiring partial decompression to identify them. As a result, TEMPEST must operate on irregular block partitions when applied directly to JPEG. Despite this challenge, TEMPEST achieves competitive performance on image data, as shown in Table 8. For reference, the method of Perez et al. Pérez et al. (2024) achieves 97% accuracy on MNIST, but relies on a much larger encoder with 87M parameters and significantly higher complexity, as it performs full attention over the entire byte sequence.

Table 8: **TEMPEST for image data.** We can use TEMPEST also in the image domain, We simple replace the byte Stream of an MP3 and apply our proposal. Despite the stark difference in encoding schemes. We achieve a hig perofrmance.

|  | 1 Token | 2 Token | 3 Token | 4 Token |
|---|---|---|---|---|
| MNIST | 86.24 | 92.63 | 95.75 | **95.79** |

## 5 CONCLUSIONS

We have introduced TEMPEST a novel approach for efficient semantic understanding of compressed multimedia data directly from their byte representations. Our method is lightweight ensemble of networks, that leverages the inherent structure of compressed byte streams to enable efficient semantic classification from a compact data representation without requiring full decoding. We demonstrated TEMPEST effectiveness across for classification tasks across 4 datasets, and 3 unique Compressed file formats. Our proposal achieves classification accuracy on par with state-of-the-art transformer models while reducing the length of these feature sequence by a factor of 3, and the size of the attention matrix by a factor of 11.

These efficiency gains are particularly significant for large-scale applications that involve processing millions of media files, in addition our approach bypasses the need for file-format decoding or raw form storage. Our finding indicate that some standard techniques (like data augmentation) are directly transferable to compressed data, moreover, there are some interesting novel properties in the compressed domain as the multi bit rate inference, or the use of multiple bit rates at training time as data augmentation. Beyond the current findings, this work lays the foundation for efficient byte-level semantic modelling of compressed files. Future directions include exploring modality-agnostic evaluations, large-scale pre-training, and further architectural optimizations to enhance efficiency.

**Reproducibility Statement**   We regard reproducibility as a priority. Upon acceptance, we will release the TEMPEST model weights, along with the full code to reproduce its training. We will include the same hyper-parameter configuration used in this paper. For experimentation we set the random seeds for PyTorch, CUDA, and python, which enforces deterministic behaviour in our experiments.

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

## A  APPENDIX

**Use of Large Language Models**  We used large language models exclusively as an styling tool, improve the manuscript vocabulary and readability, correcting grammar, and improving clarity. At no point was an LLM used to decide or influence the research design, data analysis, experiments, or the research conclusion

