# OpenReview forum: "Transformers from Compressed Representations"
_ICLR.cc/2026/Conference — Submitted to ICLR 2026_

### Official Review · Reviewer_R1ie · 2025-10-24

**Soundness:** 3
**Presentation:** 3
**Contribution:** 2
**Rating:** 4
**Confidence:** 4

**Summary:**

The paper addresses a specific challenge in low computational resource environments, where training multimedia (audio/image) classification models can be costly due to large data sequence lengths. Existing compression methods typically focus on partially decoded formats or byte-level operations and prioritize input modeling over classification. This study proposes learning directly from compressed representations at a natural block level instead of a byte level, as blocks better retain the semantic information of the data.


The authors utilize natural data blocks found in formats such as MP3, Opus, and JPEG. These blocks are embedded using a transformer network, generating embeddings for each block. Subsequently, these embeddings are compacted in sequence length using a MLP-Mixer-like network, forming the most compressed version, termed the Block Embedded Sequence. This sequence is regularized with a reconstruction objective for regularization purposes rather than as a final goal. The compressed sequence is then processed by a transformer encoder and a linear classifier to produce class predictions.


Experiments were conducted on audio formats in MP3 and Opus at varying bitrates (at least for MP3) and JPEG for images, using the ESC-50, SpeechCommandsV2, and AudioSet datasets for audio, and MNIST for images. The results demonstrate comparative performance to the Audio Spectrogram Transformer model, with reduced computational cost.

**Strengths:**

- The research demonstrates that compressed formats are viable for classification tasks, offering significantly reduced computational costs.
- Extensive ablation studies examine the impact of Block Embedded Sequence Length, Reconstruction Loss, Depth of the Block Embedding Network, bit rates, and other variables.

**Weaknesses:**

- The paper appears to focus on a relatively specialized and narrow problem area. Expanding the evaluation to include a broader range of tasks would enhance the understanding of the method's overall applicability and value.
- Additionally, the block definitions seem tailored to specific inputs, raising questions about their scalability to other formats. A comprehensive analysis in these areas would be beneficial.

**Questions:**

Could you provide any insights or analyses on the performance of intermediate sequences in tasks beyond classification? How much informational content is retained within these intermediate sequences?

---

> ### Author Response · Authors · 2025-11-24
> **Specialized and narrow problem area**
>
> We acknowledge the reviewer's observation regarding the scope of application. However, we contend that this is a consequence of the ubiquity of the transformer architecture in machine learning, rather than a shortage in our study. Testing every media type and compressed file format would be infeasible for the scope of a single manuscript.
>
> We would like to note that our primary contribution is not to achieve state-of-the-art performance across every multi-media data format. Rather, establishing the feasibility of transformers operating effectively on compressed data. To the best of our knowledge, no prior work has presented a practical and effective method for tokenization from compressed streams. Our paper's main contribution is to design the first feasible compressed data tokenizer by analyzing the structural blocks present in compressed formats, rather than pursuing a breadth of applications.
>
> We believe that demonstrating a first practical tokenization strategy can provide a foundation for future research endeavors across more diverse data domains.

---

> ### Author Response · Authors · 2025-11-24
> **Tailored Block Definitions**
>
> TEMPEST does not rely on handcrafted byte-streams or codec-specific rules. TEMPEST relies only on the existence of locally decodable units, which are a common design feature of compressed representations. We clarify that we do not define the compressed blocks or their structure, as these are already specified by their respective compressed file format (CFF) standards. Our only intervention in the CFF is selecting the compression quality by specifying a bit rate (Line 226). The default codec then constructs the compressed byte stream according to the CFF standard and our specified bit rate.
>
> For training and inference, we simply parse the file identifying the byte markers for each format, and divide the byte stream at these marker locations. We create our tokens as the sub-sequence of bytes between consecutive markers. Importantly, our networks across all 3 CFFs are identical and require no modification or tailoring to the CFF byte stream. Without any format-specific adaptation our tokenizer achieves competitive results against state-of-the-art methods using a standard transformer architecture.
>
> Finally, we emphasize that TEMPEST does not require perfect boundary detection or any semantic parsing. Our JPEG experiment uses approximate boundaries in the byte stream, and yields competitive results, demonstrating some level of robustness to imperfect block boundaries.

---

> ### Author Response · Authors · 2025-12-03
> **Intermediate sequences in tasks beyond classification.**
>
> While the main paper focuses on clip-level classification. The underlying transformer in TEMPEST’s architecture naturally produces block-level embeddings, and these sequences can be used directly for tasks requiring fine-grained temporal resolution.
>
> To evaluate the information content preserved at this block level, we conducted an additional experiment on voice activity detection (VAD) using the AVA-Speech dataset [A]. VAD is analogous to segmentation on the 1-D domain and  requires labels, making it a suitable probe of the features in the block-embedded sequence.
> We train TEMPEST’s on AVA speech from scratch and obtain 92.45% AUC, which is slightly better than the baseline in [A] of 91.7.
>
> We can not provide a direct comparison to AST as the AST network is not designed for this task, and further adaptation from its codebase is required.
>
> This result indicates that the block embeddings retain useful temporal and semantic details to support tasks that operate at a much finer granularity than clip-level classification. In short, the compression-derived tokens preserve not only global semantics but also localized, frame-level information necessary for boundary-sensitive tasks.
>
> [A]Chaudhuri, Sourish, et al. "Ava-speech: A densely labeled dataset of speech activity in movies." arXiv preprint arXiv:1808.00606.

---

### Official Review · Reviewer_X8FF · 2025-10-29

**Soundness:** 3
**Presentation:** 3
**Contribution:** 3
**Rating:** 6
**Confidence:** 3

**Summary:**

The paper introduces TEMPEST, a novel transformer-based approach that directly learns semantic representations from compressed multimedia data (MP3, Opus, JPEG) without requiring full decoding. This is achieved by leveraging the inherent block structure of these formats for natural tokenization, employing a two-stage architecture that processes block-level embeddings followed by sequence-level aggregation. TEMPEST delivers significant efficiency gains, including a 3x reduction in token sequence length and an 11x smaller attention matrix compared to traditional baselines, while maintaining competitive accuracy across diverse datasets.

**Strengths:**

1. **Novel Conceptual Framework**: The paper introduces TEMPEST, a fundamentally new paradigm to multimedia processing by working directly with compressed formats, e.g., MP3 and JPEG. The authors also incorporate innovative training techniques like bit rate augmentation and multi-bit rate inference that further improve model performance, generalization, and robustness.


2. **Practical Utility**: The demonstrated efficiency gains have clear practical applications in real-world systems with 3x reduction in token sequence length (32 tokens/sec vs. 108 tokens/sec for AST) and 11x reduction in attention matrix size (1,024 elements vs. 11,664 elements).

**Weaknesses:**

1. **Limited Baseline Comparisons:** The current evaluation primarily benchmarks TEMPEST against AST. While AST is a direct and relevant baseline, it was published in 2021, and the field has seen significant advancements since then.  To fully contextualize TEMPEST's contributions and thoroughly assess its performance, it would be beneficial to include comparisons with more recent methods in the relevant domain.

2. **Dataset Complexity:** For the image experiment, the evaluation largely relies on datasets such as MNIST, which are relatively simple and may not fully reflect the challenges of real-world scenarios. The authors should consider evaluating TEMPEST on more complex datasets to better demonstrate the method's effectiveness. For instance, despite potential computational resource constraints, exploring performance on CIFAR-10 or CIFAR-100 would offer stronger evidence of the method's robustness and generalizability.

3. **Out-of-Distribution Capability:** The current experiments mainly focus on classification tasks and do not explicitly address TEMPEST's performance under out-of-distribution conditions. However,  investigating how TEMPEST handles or identifies OOD samples would significantly strengthen the paper's claims regarding its practical utility.

4. **Format Dependency and Unification**: The paper mentions that TEMPEST is applicable across various data formats (e.g., MP3, JPEG). However, it appears that the method still exhibits a degree of format dependency, requiring separate training for each format. Could the authors elaborate on whether it is possible to train a single, unified model of TEMPEST that can simultaneously process and handle multiple distinct data formats without format-specific adjustments?

**Questions:**

The proposed TEMPEST appears to offer a promising approach for efficiently processing large-scale data. Given this potential, it would be beneficial for the authors to discuss how TEMPEST relates to and can potentially combine with existing "adaptive token methods" (e.g., those that dynamically adjust sequence length or token importance [1,2,3]) in the conclusion.

[1] Duggal, Shivam, et al. "Adaptive length image tokenization via recurrent allocation." First Workshop on Scalable Optimization for Efficient and Adaptive Foundation Models. 2024.

[2] Wen, Tiansheng, et al. "Beyond Matryoshka: Revisiting Sparse Coding for Adaptive Representation." Forty-second International Conference on Machine Learning.

[3] Hu, Wenbo, et al. "Matryoshka query transformer for large vision-language models." Advances in Neural Information Processing Systems 37 (2024): 50168

---

> ### Author Response · Authors · 2025-11-25
> **Limited Baseline Comparisons:**
>
> We acknowledge that including additional baselines could provide further contextualization for TEMPEST's performance. However, our selection of AST as the primary baseline was motivated by two considerations:
>
> (i) Direct architectural comparability: AST represents the most direct and fair baseline for our evaluation, as it relies exclusively on the transformer architecture and self-attention mechanisms to classify audio data, our approach uses the exact same tools. The only architectural differences between AST and TEMPEST are the components specifically designed for compressed data (the reconstruction network and block embedding encoder) and the compressed input representation. This makes AST the most appropriate baseline for isolating and evaluating the effectiveness of our proposed components. By comparing against AST, we can directly demonstrate the advantages of operating in the compressed domain while controlling for architectural variations.
>
> (ii) Limited compressed-domain baselines: Compressed-domain methods remain scarce in the literature. With the exception ofr Horton et al. (2023a) abd  Pérez et al. (2024), most recent approaches operate on raw waveforms or spectrogram representations, making direct comparisons far less meaningful for evaluating compressed-domain data.
>
> Furthermore, more recent methods typically incorporate augmented transformer architectures, task-specific modifications, or multi-modal frameworks that introduce orthogonal contributions beyond TEMPEST's core focus. Including such baselines would generate confusion in the attribution of the empirical gains, as performance discrepancies could be attributed to the fact that tempest operates in the compressed domain or the lack of additional mechanism built on the AST architecture.
>
> We believe our baseline selection provides the most direct and controlled evaluation of TEMPEST's contribution. However, we are open to including additional context about related methods in the revised manuscript if the reviewer finds this valuable.

---

> ### Author Response · Authors · 2025-11-26
> **Comparison to adaptive token methods**
>
> We appreciate the suggestion to discuss adaptive tokenization methods, and acknowledge there are some conceptual similarities with our work. However, we note one important distinction between the approaches presented in [1–3] and TEMPEST.
>
> The referred works [1–3] approach token efficiency by operating on high-level semantic features extracted, typically, after a full pass on the visual encoder. In contrast, TEMPEST focuses on mapping byte-level inputs, these inputs are defined by the compressed file format (CFF), and are devised to be efficient for disk storage. Critically, individual bytes in a CFF carry no inherent semantic information, whereas the methods in [1–3] explicitly rely on semantic content to guide their sequence adaptation strategies.
>
> While both approaches aim to reduce sequence length, we view the works in [1–3] as largely orthogonal to our contribution. TEMPEST addresses the challenge of creating a practical tokenizer for compressed-domain data, whereas adaptive tokenization methods optimize the processing of tokens with semantic meaning.
>
> To augment our work, we will add a discussion in the revised manuscript contextualizing our work relative to adaptive tokenization methods and outlining potential avenues for combining these complementary approaches. We emphasize that our core contribution remains the development of a practical and effective tokenizer that enables feature learning directly from CFFs.

---

> ### Author Response · Authors · 2025-12-03
> **Out of Distribution Capabilities.**
>
> This is an excellent question, we appreciate the opportunity to further clarify this important aspect of TEMPEST. We begin by referring the reviewer to Tables 5 and 6, which describe our training and inference configurations, respectively. Table 5 specifies that our training scheme employs three bit rates: [20, 26, 32] kbps. In contrast, Table 6 demonstrates that our multi-frequency inference pipeline has four bit rates: [23, 26, 29, 32] kbps.
>
> The bit rates of 23 kbps and 29 kbps are not included in our training set, yet they contribute to test-time performance (about 1%). This is noteworthy, as stated in Line 232, varying the bit rate alters about 90% of the byte values in the block encoding. The bit rates of 23 kbps and 29 kbps therefore represent out-of-distribution input samples.
>
> This result suggests that TEMPEST learns representations that are resilient to variations in compression quality, and the low level byte values. Rather than overfitting to the specific bit rates used during training.
>
> To further validate this empirical result, we have designed additional experiments. We train over 2 of our bit rates, and validate exclusively on the third one. On the second experiment we evaluate
>
> | Training Bit rates (kbps)| Validation Bit rate (kbps)| Accuracy |
> |------------|-------|-------|
> | 20, 26, 32 | 32 | 91.27 |
> | 20, 32  | 26 | 90.84 |
> | 20, 26  | 32 | 91.15 |
> | 26, 32  | 20 | 90.89 |
>
> Overall we observe a slightly lower than our baseline (-0.43). However, the small variation appears to be more correlated  with the validation bit rate (best validation result is 91.15 for 32kbps) rather than the actual training setup.

---

> ### Author Response · Authors · 2025-12-03
> **Format Dependency.**
>
> While our original submission makes no claims regarding cross-format interoperability within a single unified model, TEMPEST's architectural uniformity enables joint training across CFFs without modification. To validate this capability, we conducted an experiment training on the ESC-50 dataset while alternating between two different CFFs within each training batch: MP3 and Opus. This cross-format training approach allows us to assess whether a single model can learn effective representations from multiple compression formats simultaneously. The following table summarizes our results.
>
>
> | Training | Inference | Split 1 | Split 2 | Split 3 | Split 4 | Split 5 | Average |
> |-------|-------|-------|-------|-------|-------|-------|-------|
> |MP3		| MP3 	| 56.52 |57.06 	| 58.96 | 58.69 | 63.68 | 58.98 |
> |MP3 +OPUS 	| MP3 	| 60.68 | 55.16 | 59.24 | 60.70	| 60.40	| 58.82 |
> |Opus 		| Opus | 48.69 | 49.42	| 51.51 | 45.45 | 51.76	| 49.36 |
> |MP3 +OPUS 	| Opus 	| 47.01 | 48.28 | 51.63 | 50.0	| 48.84 | 48.63 |
>
>
> We observe a minimal  performance gap for Opus (−0.73%) and a slight increase for MP3 (+0.16%). The magnitude for either variation remains small, suggesting that TEMPEST's tokenization mechanism can effectively learn representations from multiple CFFs simultaneously without any relevant interference between formats.
> This result further validates that our approach is not inherently limited to single-format scenarios and also supports that no individual tailoring is needed for multiple CFFs. We will include this analysis in the revised manuscript to clarify TEMPEST's potential for multi-format learning.

---

### Official Review · Reviewer_q9rc · 2025-10-30

**Soundness:** 3
**Presentation:** 2
**Contribution:** 3
**Rating:** 4
**Confidence:** 2

**Summary:**

This paper proposes a compressed-domain transformer framework for efficient semantic representation learning named TEMPEST. Specifically, TEMPEST includes two core modules: Block Embedding Network and Classification Network, regularized by a Block Reconstruction Network. The Block Embedding Network leverages the intrinsic block structure of various compressed file formats to tokenize byte streams into blocks, applies intra-block attention with lightweight transformers, and compresses them into embeddings. The Classification Network aggregates these block embeddings across an entire file using a standard transformer encoder with a [CLS] token for downstream classification. The Block Reconstruction Network guides the embedding process via a reconstruction loss, ensuring semantic fidelity and compactness without full media decoding.

The experimental evaluation in this paper assessed the performance of the proposed TEMPEST across audio and image domains, including three compressed formats and multiple datasets, comparing it with state-of-the-art baselines. The results indicate that the proposed TEMPEST achieves competitive or superior accuracy while reducing sequence length and attention matrix size, thereby providing computational and memory efficiency gains.

**Strengths:**

- Directly embedding features from compressed files instead of raw data avoids the additional storage and transmission overhead introduced by the decoding process, which is a meaningful and practical advantage for real-world applications.
- Experimental results demonstrate that TEMPEST can improve efficiency while maintaining competitive performance.

**Weaknesses:**

- The method relies heavily on structural characteristics of specific compression formats, which limits its generality. TEMPEST’s core idea is to use compression blocks rather than bytes as token units, which requires the compression format to have explicit and easily parsable minimal independent decoding units. The authors may need to explicitly clarify which compression formats are supported and whether extra engineering adaptation is required for different compression algorithms.
- The experiments primarily evaluate on MP3, Opus, and JPEG, lacking verification on other common formats such as H.264, HEVC, and FLAC. There is also no analysis of the relationship between different compression formats and model performance.
- Although TEMPEST shows advantages over AST in FLOPs and TPS, the design requires coupling between different sub-networks, making the method more complex, and lacking corresponding experimental analysis.
- The experiments are mostly conducted on small-scale datasets. It remains unverified whether compressed-domain training with TEMPEST on large-scale data (e.g., compressing ImageNet and training with TEMPEST) can match the multi-task generalization performance of raw-domain training.
- The audio tasks are limited to classification, with no evaluation on other task types such as detection, segmentation, or generation.

**Questions:**

Please refer to weaknesses.

---

> ### Author Response · Authors · 2025-11-25
> **Coupling and analysis of components.**
>
> We respectfully disagree with the reviewer's assessment of added complexity. While TEMPEST introduces two additional modules (the block-embedding encoder and the reconstruction network), their designs are lightweight and their interactions are straightforward. In terms of component overhead, TEMPEST adds exactly two linear layers (the channel mixer) to the base architecture (ViT) at inference time. At training time, the reconstruction network introduces only two additional layers.
> The block embedding encoder is a standard two-layer transformer encoder which introduces no additional architectural complexity with respect to the base architecture (ViT). Its only purpose is to take the compressed block and yield a much shorter sequence which is done with the channel-mixer.
> The reconstruction network is even more light-weight, it is composed of a single transformer decoder layer, and one linear layer for prediction. Its only purpose is to regularize the training through the reconstruction loss (L_r) and it is entirely discarded at inference time.
> In Section 4.3, we provide a detailed ablation study examining each component of the TEMPEST architecture, including: (1) the depth of the reconstruction network, (2) the effect of the reconstruction loss, (3) a fully end-to-end transformer architecture that replaces the separate encoder and (4) the sensitivity analysis to the hyper parameter L’. We think that these ablations demonstrate the contribution of each design choice to overall performance.

---

> ### Author Response · Authors · 2025-11-25
> **Reliance on structural characteristics of the compression format.**
>
> Our experimental validation demonstrates that TEMPEST operates effectively across MP3, Opus, and JPEG formats. In each case, TEMPEST relies on two basic assumptions about the compressed format: (i) locally independently decodable blocks, and (ii) approximate segments using format-provided markers. While these assumptions are satisfied by a significant number of codecs across different media types, exhaustively analyzing every codec for every media format would be infeasible and beyond the scope of a conference paper. We will revise the manuscript to explicitly enumerate the codecs investigated and clarify the generalizability of our approach.
> Although we can not validate in every single CFF, we emphasize that our primary contribution is not to benchmark performance across every compressed multimedia format. Rather, we aim to establish the fundamental feasibility of transformers operating directly on compressed data and the efficiency benefits derived from it. Our work provides empirical evidence across representative audio and image formats, establishing a foundation for future research to explore additional codecs and modalities.
> Finally, we note that TEMPEST does not require perfect sub-block matching. Our JPEG results use approximate MCU boundaries and the model learns effectively, showing that TEMPEST only requires locally coherent segments rather than bit-level alignment with the compressed block.

---

> ### Author Response · Authors · 2025-12-03
> **Aditional Audio Tasks**
>
> While the main paper focuses on clip-level classification. The underlying transformer in TEMPEST’s architecture naturally produces block-level embeddings, and these sequences can be used directly for tasks requiring fine-grained temporal resolution.
>
> We leverage the information content preserved at this block level, we conducted an additional experiment on voice activity detection (VAD) using the AVA-Speech dataset [A]. VAD is analogous to segmentation on the 1-D domain and  requires labels, making it a suitable probe of the features in the block-embedded sequence.
>
> We train TEMPEST’s on AVA speech from scratch and obtain 92.45% AUC, which is slightly better than the baseline in [A] of 91.7.
>
> This result indicates that the block embeddings retain useful temporal and semantic details to support tasks that operate at a much finer granularity than clip-level classification. In short, the compression-derived tokens preserve not only global semantics but also localized, frame-level information necessary for boundary-sensitive tasks.
>
> [A]Chaudhuri, Sourish, et al. "Ava-speech: A densely labeled dataset of speech activity in movies." arXiv preprint arXiv:1808.00606.

---

### Official Review · Reviewer_bCZR · 2025-11-01

**Soundness:** 3
**Presentation:** 4
**Contribution:** 3
**Rating:** 6
**Confidence:** 2

**Summary:**

This paper proposes TEMPEST, a tokenizer+encoder pipeline that learns directly from compressed file formats (CFFs) by treating blocks (e.g., MP3 frames, JPEG MCUs, Opus frames) as atomic units rather than bytes. Each compressed block is embedded with a lightweight intra-block transformer and channel mixer, then a vanilla transformer encoder aggregates block embeddings for classification. Across ESC-50, SC2, and AudioSet, TEMPEST is competitive with AST while using far fewer tokens per second and lower FLOPs.

**Strengths:**

1. The paper’s central idea—treating self-contained codec blocks as tokens—is clean and broadly appealing, because it avoids byte-boundary misalignment while preserving the semantics that matter.
2. Despite its simplicity, the approach delivers strong efficiency: it cuts sequence length and attention cost substantially yet remains competitive in accuracy, especially on data-limited regimes like ESC-50.
3. The small reconstruction head is a practical touch that consistently sharpens the block embeddings. I also appreciate the codec-aware perspective on augmentation: training and even inference with multiple bit rates act like multi-view learning and yield measurable gains.
4. Finally, the method shows encouraging performance with a lightweight architecture, suggesting the idea travels beyond a single codec or modality.

**Weaknesses:**

1. The approach assumes block-structured formats with discoverable boundaries. Not all compression schemes expose clean block markers (even JPEG MCU boundaries require approximation), which may limit generality and complicate deployment outside MP3/Opus/JPEG.
2. TEMPEST trails AST on SC2 and AudioSet at ~65% FLOPs and much shorter sequences; the paper doesn’t explore accuracy–compute scaling laws (e.g., what happens if TEMPEST matches AST’s FLOPs, tokens, or parameters?).
3. Ablations show accuracy benefits from deeper block embedding encoders but also increased FLOPs (since Et runs over length-144 intra-block tokens). It’s unclear, at fixed FLOPs, whether adding capacity to Et (embedding) or Ec (classifier) yields better returns.

**Questions:**

1. How does TEMPEST scale in FLOPS compared to AST —does accuracy close the gap on SC2/AudioSet if you increase depth/width or raise?

2. Under a fixed FLOPs budget, where should capacity go for the best returns—into the block encoder E_t or the sequence encoder E_c.

3. For codecs without clean, explicit block markers, can you learn or detect boundaries reliably and how does boundary error impact accuracy?​

---

> ### Author Response · Authors · 2025-11-24
> **Discoverable boundaries and lack of generality**
>
> We acknowledge that CFFs differ in how explicitly they expose block boundaries. However our experimental setup suggests that TEMPEST does not require perfect or byte-exact boundaries. Our JPEG experiment uses approximate MCU boundaries, yet TEMPEST still performs well under these conditions. Importantly, we note that TEMPEST does not rely on format-specific semantics, it only requires a notion of compressed sub-components, which are present across several practical codecs. We will revisit the paper to state this more clearly.
> Regarding the generality and deployment, we note that the MP3, Opus, and JPEG compressed file formats already encompass audio and image data, which are the two most prevalent types of multimedia. We do not consider our reliance on block segments a limitation in generality, as data in compressed formats that cannot be readily decomposed into blocks (e.g., JPEG 2000) can always be converted to block-based formats (e.g., JPEG) through standard transformation pipelines.
> This practical constraint is analogous to the practical requirements for the input data in standard transformer architectures. A transformer would impose specific input constraints: a particular channel format (RGB, BGR or single channel, for spectrograms), a dataset-dependent mean subtraction during preprocessing, and a fixed tokenization patch sizes (e.g., a transformer trained with 8×8 patches cannot process 16×16 tokens without significant input modifications). Our approach operates within comparable practical boundaries that are well-established in both the vision and compressed-domain literature.
> We kindly invite the reviewer to reassess the identified limitations, as they are inherent to working with structured data format, and naturally exist in every domain.

---

> ### Author Response · Authors · 2025-11-26
> **Scaling Tempest  on SC2**
>
> We conducted additional experiments with a model containing double the parameter count of the original TEMPEST.  For simplicity, we kept the number of layers constant and scaled the number of parameters of the linear layers. This scaling choice brings an additional question: how should additional parameters be allocated across the architecture?
>
> We evaluate three possible configurations:
> TEMPEST_e: Additional parameters allocated exclusively to the block embedding sub-network
> TEMPEST_c: Additional parameters allocated exclusively to the classification sub-network
> TEMPEST_s: Additional parameters distributed across both sub-networks
>
> For each configuration, we evaluated performance at 32 kbps inference and the four-frequency inference setting from Table 6. We performed hyperparameter tuning for learning rate, CutMix parameters, and dropout in the classification sub-network. The results are presented below:
>
>
> | Model      | Single  | Multi  | FLOPs |
> |------------|-------|-------|--------|
> | AST        | 92.6  | N/A   | 25.65G |
> | TEMPEST    | 91.27 | 92.32 | 16.85G |
> | TEMPEST_e  | 90.99 | 91.68 | 64.89G |
> | TEMPEST_c  | 91.8  | **92.66** | 17.60G |
> | TEMPEST_s  | 90.99 | 92.29 | 40.01G |
>
>
> We make the following observations:
> Allocating additional parameters exclusively to the block embedding sub-network (TEMPEST_e) provides no empirical benefit and incurs a substantial computational penalty (64.89G FLOPs).
> A balanced parameter allocation (TEMPEST_s) results in slightly diminished performance while also introducing a large computational overhead (40.01G FLOPs).
> Allocating parameters to the classification sub-network (TEMPEST_c) yields improved performance, with multi-frequency inference (92.66%) now surpassing AST (92.6%) by a small margin. Notably, TEMPEST_c introduces a minimal computational overhead of 0.75G FLOPs, maintaining lower complexity compared to AST (25.65G FLOPs).
> These findings suggest our  block embedding sub-network is compatible with larger classification sub-networks which benefit from the reduced token length sequence. As the classification network does not deal with any compressed data and is composed exclusively of vanilla transformer layers, we hypothesize that this component follows standard scaling laws. We will include these results in the revised manuscript.

---

> ### Author Response · Authors · 2025-11-26
> **FLOP allocation analysis.**
>
> This is an excellent question. However, maintaining an exact FLOP budget of 16.85G while varying the architectural depth (as in Table 4) requires adjusting the layer size in the  sub-networks E_t and E_c, even then, a precise FLOP matching is not achievable.
> To address this concern, we design an experiment with two additional configurations (TEMPEST_1 and TEMPEST_3) that correspond to the additional architectures from Table 4 (first and last rows). Both models approximate our baseline FLOP count: TEMPEST_1 has 16.89 GFLOPs (+0.04G) and TEMPEST_3 has 17.26 GFLOPs (+0.41G).
> To achieve these FLOP targets, we adjusted all linear layer dimensions to 258 for TEMPEST_1 and 192 for TEMPEST_3. We note that these adjustments change the parameter counts as follows:
> TEMPEST_1: +3M additional parameters
> TEMPEST_3: −1.1M fewer parameters
> For both models, we performed learning rate tuning to ensure fair comparisons. The results are presented below:
>
>
> | Model      | Col1  | Col2  | FLOPs |
> |------------|-------|-------|--------|
> | TEMPEST    | 91.27 | 92.32 | 16.85G |
> | TEMPEST_1  | 91.53 |92.4 | 16.89G |
> | TEMPEST_3  | 91.07| 92.11 | 17.26G |
>
> Analysis: TEMPEST_1 shows a slight performance improvement over the baseline, while TEMPEST_3 exhibits slight decrease in performance. However, when contextualized with our parameter scaling analysis (see response “Scaling TEMPEST on SC2”), these results suggest that performance differences are better attributed to parameter count rather than the FLOP allocation. TEMPEST_1's improvement aligns with its increased capacity (+3M parameters), while TEMPEST_3's decline corresponds to its reduced capacity (−1.1M parameters).
> This analysis indicates that for TEMPEST, the parameter count contributes the most to performance.

---

> ### Author Response · Authors · 2025-12-02
> **Boundary Sensitivity.**
>
> As discussed in our previous response: "Discoverable boundaries and lack of generality," the boundaries in JPEG encoding are always approximate, as MCUs (Minimum Coded Units) are defined at the bit level rather than byte-aligned positions.
> To provide a more in depth analysis of TEMPEST's sensitivity to boundary imprecisions, we design two additional experimental setups that deliberately introduce jitter around the block boundaries: First, we used a model pretrained on SpeechCommands v2 with exact MP3 boundaries and evaluate its performance on jittered boundaries at inference time. Second, we trained a model directly on jittered boundaries (+-N Bytes) and evaluated on the same shifted compressed blocks of our first experiment.
> For each individual block we jitter each boundary (beginning and end) by +-N bytes. Setting the jitter to 2 would result in 2 bytes from the beginning being lost, and 2 additional bytes from the next frame being included. Setting the jitter to -2 would include 2 bytes from the previous block and 2 bytes at the end being lost.
>
>
> | Jitter (Bytes) | Percentage of original bytes | TEMPEST  | TEMPEST + jitter training |
> |------------|-------|-------|-------|
> | 0     | 100 | 91.27 | 91.27 |
> | +-1  | 98.61 | 91.22 | 91.28 |
> | +-2  | 97.22 | 82.95  | 91.03 |
> | +-5  | 93.06 | 49.50 | 90.33 |
> | +-10 | 86.11 | 24.02  | 84.00 |
> | +-14 | 80.56 | 17.16 | 80.21 |
> | +-21 | 70.83 | 11.33 |76.74 |
> | +-28 | 61.11 | 8.62 | 23.99 |
>
>
> Overall we observe that baseline TEMPEST is highly sensitive to imprecise boundaries, with a significant performance penalty when only 3% of the block segment differ.  In contrast, the performance of TEMPEST trained with jittered segments directly correlates with the remaining percentage of original bytes from the block. No significant performance loss (less than 1 point) is observed if the boundary selection retains about 93% of the original window. Additional degradation is present if less than 90% of the window is preserved, and seems to directly correlate with the performance. Only when about 40% of the block original data is lost, we observe a major drop in performance.
> These results show that the original TEMPEST is sensitive to small boundary errors. However its robustness can be significantly  improved at training time without any additional training data or regularization. This experimental validation supports our central claim that meaningful representation learning from compressed data requires identifying and modelling the internal structural boundaries defined by the CFF. We will include this analysis in the final version of the paper.

---

### Meta-Review · Area_Chair_5Cn2 · 2025-12-22

**Summary:**

Reviewer bCZR, Reviewer q9rc and Reviewer R1ie pointed out not all compression data can provide block-structured formats with discoverable boundaries for the proposed method. Reviewer q9rc was concerned about the effectiveness on more general compression formats. Reviewer X8FF complained about out-of-dated compared baseline and the effectiveness on more complex image datasets. Reviewer R1ie argued that the application of the proposed method is somewhat narrow and specialized.

**Reviewer Concerns:**

The concerns about the limited effectiveness on various compression formats and more complex image datasets is not well addressed. The comparison to recent related work is also missing.

**Reviewer Scores:**

Reviewer bCZR: 6;

Reviewer q9rc: 4;

Reviewer X8FF: 6;

Reviewer R1ie: 4

---

### Decision · Program_Chairs · 2026-01-26

Reject